# Lunar Cycle, Climate, and Onset of Parturition in Domestic Dromedary Camels: Implications of Species-Specific Metabolic Economy and Social Ecology

**DOI:** 10.3390/biology12040607

**Published:** 2023-04-17

**Authors:** Carlos Iglesias Pastrana, Francisco Javier Navas González, Juan Vicente Delgado Bermejo, Elena Ciani

**Affiliations:** 1Department of Genetics, Faculty of Veterinary Sciences, University of Córdoba, 14014 Córdoba, Spain; 2Department of Biosciences, Biotechnologies and Environment, University of Bari ‘Aldo Moro’, 70125 Bari, Italy

**Keywords:** labor onset, lunar phase, weather factors, offspring sex, social communication, heterothermia, epigenetic control

## Abstract

**Simple Summary:**

Despite traditional reports on the ability of female dromedaries to modulate pregnancy length in response to environmental conditions, no empirical study has been developed. According to the present results, female dromedaries would adjust the onset of parturition to give birth around darker and, therefore, safer nights since social communication and interaction between congeners are increased under dim light conditions, mostly mediated by hormonal signals. Furthermore, the time of delivery, a highly energy-demanding process, may also be modulated by mean wind speed and its transient increases since these weather variables affect the thermal comfort and, thus, the heat transfer between an animal and its environment and the individual energy budget. Gravid females may be more energetically compromised when the offspring is a male; hence, the onset of parturition for the newborns of the male sex will be more probable to occur on slightly brighter nights and when mean wind speed is lower when compared to their female counterparts. Such specific phenology would favor a proper multisensory interaction between the mother and the more immature male young at neonatal stages so that thermoregulatory demands are minimized for these reproductive tasks, and then the fitness of parents and the survival of offspring are improved.

**Abstract:**

Given energy costs for gestating and caring for male offspring are higher than those of female newborns, external environmental conditions might be regarded as likely to affect the timing of delivery processes differentially depending on the sex of the newborn calf to be delivered. The aim of the present paper is to evaluate the association between environmental stressors such as the moon phase and weather-related factors and the onset of labor in female dromedaries. A binary logistic regression model was developed to find the most parsimonious set of variables that are most effective in predicting the probability for a gravid female dromedary to give birth to a male or a female calf, assuming that higher gestational costs and longer labor times are ascribed to the production of a male offspring. Although the differences in the quantitative distribution of spontaneous onset of labor across lunar phases and the mean climate per onset event along the whole study period were deemed nonsignificant (*p* > 0.05), a non-negligible prediction effect of a new moon, mean wind speed and maximum wind gust was present. At slightly brighter nights and lower mean wind speeds, a calf is more likely to be male. This microevolutionary response to the external environment may have been driven by physiological and behavioral adaptation of metabolic economy and social ecology to give birth to cooperative groups with the best possible reduction of thermoregulatory demands. Model performance indexes then highlighted the heterothermic character of camels to greatly minimize the impact of the external environment. The overall results will also enrich the general knowledge of the interplay between homeostasis and arid and semi-arid environments.

## 1. Introduction

Circadian rhythm, or the physiological functions and behavioral responses that are regulated in a 24-h day-night cycle, have been widely investigated in an attempt to understand its adaptive role, evolution, and relative significance [1,2]. Photoperiod is indeed the most cited factor within the theoretical background and analyses of the environmental cues responsible for oscillations on circadian timing [3,4,5]. The secretion of the hormone melatonin, being inversely related to day length, provides organisms with the signal to measure time physiologically and thus maximize the investment of energy differentially among competing processes, ranging from subcellular metabolism to more complex traits such as thermoregulation and behavior [6]. In the last case, although many behaviors affected by photoperiod are associated with reproduction seasonality, some affective responses, cognitive outputs, and migratory states are also controlled by environmental light independently of circulating gonadal steroids [7,8,9].

The exploration of the effects of the lunar cycle on animal behavior and physiology represents the most striking example of how not only hormonal but also immunological and neural pathways that underlie complex phenotypic traits are modulated by the amount of light and darkness in a daily cycle. Being the putative molecular mechanisms that are still undetermined, moonlight intensity, geomagnetism, and gravity are argued to be more likely the primary causes of the periodic deviation in frequency or intensity for specific brain and body functions. Specifically, defense behavior and spacing, predator-prey dynamics [10,11], and systematic variations in birth rate, fertility, ovulation, and locomotor activity, are reported to be lunar-synchronized events in different species and that have particular adaptive values depending on the ecological niche of each aggrupation of organisms [12,13,14,15,16].

In spite of the fact that the light–darkness cycle may be the primary environmental cue used by living organisms to synchronize physiological processes with regular exogenous fluctuations, this temporal rhythm is also sensible to varying weather, so the circadian clock has evolved to shape both the dynamics of the light–darkness cycle and climatic signals [17,18]. Warm, dry, and calm conditions are linked to the advancement of the timing of reproduction and the date of termination of hibernation and a decrease in growth rate and semen quantity and quality variables in some animal species [19,20,21]. Similarly, significant associations between weather variables such as temperature, humidity, precipitation, or wind patterns and adverse pregnancy outcomes in humans [22] and cows [23] have been found. Moreover, changes in atmospheric pressure have been reported to influence the occurrence of the spontaneous onset of labor in term pregnancies and the birth rate [24,25]. However, the significance of the effects of the moon phase and weather conditions on physiology is a subject that arouses controversy within the scientific community, given the disparity of the results found. Several studies that have been carried out with regard to this applied field have concluded the effects were not statistically significant or the modeled size of the effect was not consistent enough so as for a clinical implication to be determined [26,27].

Among the most criticized errors of the aforementioned studies is the common preference for birth rate rather than the onset of labor as the variable to be correlated with the moon phase or meteorological conditions. Stern, et al. [28] stated that if the moon phase influences the birth rate, it could be only a final consequence of the effects of the satellite on the time of the onset of labor ending in the birth itself. Among the principal factors, particularly endocrine, involved in the process of animal parturition are the changes in hormonal levels of estrogen, progesterone, prostaglandins, oxytocin, corticotrophin-releasing hormone, and cortisol [29]. Although different environmental stressors are reported to interfere with the endocrinology of reproductive function [30,31,32], the exact effects of the moon cycle and climate-related factors on the mechanisms and mediation of endocrine but also nervous (e.g., allostatic load) [33] and immune systems (e.g., cytokines and surfactant proteins) [34] that lead to the initiation of parturition, are hardly elucidated for several species.

Additionally, the methods, when used, of determining individual exposure to the moon and weather effects may generate bias in the power of statistical tests to detect significant relationships [35]. Apart from that, from a meta-analytical viewpoint, the disparity of results might be a reflection of the gene variation of the circadian clock across large geographical areas, hence the latitudinal clines discovered in its gene frequencies [36]. Lastly, and strongly related to the previous argument, the inclusion of non-native organisms into the study sample could derive from the lack of strong association between local environmental conditions and the body functioning of inhabiting beings, especially if the history of adaptation dates recently and the heritability of adaptive traits remains low [37].

Canarian dromedaries are an endangered, unique breed with a historical record of local adaptation and evolution for six centuries but which lacks a rigorous estimation and control of its reproductive timing for management and conservation purposes [38]. Although an increasing tendency for intensification of camel farming systems is patent, these animals are mostly reared in traditional extensive forms [39], thus being relatively exposed to environmental stressors (climatic, chemical, wildfire, physical, and biological stressors). In these regards, environmental-based epigenetic modifications can occur in embryonic tissues and cells via mother-fetus communication through the placenta [6,40]. Furthermore, higher gestational costs and longer labor times are ascribed to the production of male offspring in comparison with female offspring [41,42]. Moreover, metabolic maturity, general vigor, and the ability to thermoregulate female offspring are expected to be reached sooner than the respective for male offspring [43]. In this scenario, dromedaries are well known to have developed exceptional physiological and biochemical adaptations to extreme arid environments and limitation of resources, in addition to the seasonal nature of their reproduction, that allow them to optimize both the utilization and storage of energy and water with a central role of thermoregulatory mechanisms [44]. Among such adaptations, female dromedaries have been reported to modulate pregnancy length in response to some environmental stresses [45,46].

Therefore, considering that parturition is a highly energy-demanding process [47] and gravid dromedary females would seek to maintain homeostasis, different impacts of environmental factors in the onset of labor depending on the sex of offspring could be expected given the abovementioned differential sex-mediated energy costs. That is an adaptive allocation of energy to maximize the constancy of the internal environment and general fitness. In this context, the present research aims to study the effects of the lunar cycle and weather conditions on the spontaneous onset of labor, depending on the sex of the offspring, in Canarian dromedary camels.

To the best of our knowledge, this is the first approach to the potential prediction of the spontaneous onset of labor based on a large dataset of environmental cues in dromedary camels, which includes additional weather factors different from temperature, relative humidity, and length of daylight. From the perspective of clinicians and breeders, in the absence of dedicated algorithms and machine-learning technologies for calving prediction in this animal species, these results would be of help in a meaningful manner to know which factors impact the most on the onset of parturition and, thus, plan adapted perinatal assistance to favor the establishment of the mother-calf bond in the best conditions, which is crucial to stimulate milk let-down in these animals [48]. In addition, the research community will be provided with a deeper empirical exploration of phenotypic plasticity in dromedaries and boosted for the examination of both the existence and extent of latitudinal clines in circadian clock gene frequencies in these animals, in view of the fact that their historical range of habitat is relatively restricted [49]. Altogether, such outcoming pieces of knowledge will assist in filling knowledge gaps on physiological ecology and ecological genetics in camels [50].

## 2. Materials and Methods

### 2.1. Animal Sample: Candidate Selection

The data used for the present retrospective study resulted from the registries of the date and hour of initiation of 415 spontaneous onsets of labor at term over the period of time between January 1992 and March 2022, both inclusive, in Canarian she-dromedaries reared at Fuerteventura (Canary Islands, Spain; 28°25′57″ N–14°00′11″ W). Concerning the sex of the resultant offspring, the study sample comprised 208 male and 207 female newborns. Each calving event registered concerned a single different gravid primiparous female; hence no subsequent parturitions of the same gravid she-dromedary existed in the database. This way, the reproductive phenology across the lunar cycle and weather in the study population can be evaluated under the scope of epigenetics, that is, inherited modifications linked to the exposure to environmental factors and that are not patent in DNA sequence [51].

Initially, gravid she-dromedaries were pre-selected based on a clinical examination to ensure the proper health status of the pair dam-fetus at the final stage (last month) of the normal gestation period. Then, the clinical signs that were subjected to observation in the pre-selected gravid females as indicators of the spontaneous initiation of labor were the following: relative isolation from the main herd, restlessness with the alternate of lying down and standing up, increase in urination frequency, and swelling of the vulva, udder, and teats [52]. When at least three of these symptoms were observed simultaneously in a gravid female, the date and hour of the observation were annotated as the spontaneous onset of labor. The reliability of these clinical signs to advance the time of parturition was confirmed with the appearance of the first water bag at the vulva within the next few minutes. Once the calf was upright with all four legs fully extended for the first time, the sex of the newborn (response variable) was recorded. All the episodes of the spontaneous onset of parturition recorded for this study were those that ended in the successful survival of the offspring.

All calving events used for this investigation were produced at the same facilities, without any prominent variation in housing conditions during the study period to reduce the magnitude to which the effects of environmental variables on the onset of parturition could be masked by other factors both on an individual camel and herd level.

### 2.2. Farming Environment: A Historical Background

The study site (Fuerteventura, Canary Islands, Spain) is the most representative emplacement where Canarian dromedaries, the only camel breed along Europe and cataloged in danger of extinction by the Spanish Ministry of Agriculture, Food, and Environment since 2012 (Order AAA/251/2012), have been and still are bred and used for the exportation of live animals. For utilitarian and managerial reasons, Canarian dromedaries are reared in single-sex groups with the best procured homogeneous distribution of age structure [53].

Since their arrival to the archipelago from the nearest African coast circa 1405, dromedaries in the Canary Islands mainly served as service animals for rural laborers and were immersed in family farming production regimes in which a lack of rigorous genealogical records was routinely practiced [54]. However, the mechanization of these activities from the last third of the twentieth century led to a decrease in the census and obligated the local breeders to seek new functional niches for their animals. Fortunately, the relatively rapid conversion of dromedaries into a far-iconic attractiveness within the touristic and recreational industry in the archipelago around the 1990s resulted in a population recovery [55]. Thenceforth, Canarian dromedaries were progressively allocated at higher stocking densities in more controlled environments to carefully manage populations for their reproductive and genetic health [54].

The housing environment of the study population consisted of square-shaped fenced corrals with a shelter creating a shaded area in the middle of the pen that had enough extension so that the total number of dromedary camels kept in the same pen could lie on it. Both the feeding and drinking points were placed along one of the lateral sides of the facility.

### 2.3. Moon Phase and Weather Variables: Data Collection

In addition to the response variable (sex of the calf), the records were completed with the moon phase and the average value for weather variables during the 24 h previous to the registered hour of onset of parturition for each gravid female. This timeframe is reported to be the most reliable temporal period to detect signals that are significantly related to the date of delivery [56].

Moon phase information was retrieved from lunar calendars. The number of days since the previous full moon was calculated for each onset of labor date, following the methodology of Grant, Halliday and Chadwick [10]. Eight moon phases were analyzed: full moon to waning gibbous (Lunar phase 1), waning gibbous to the third quarter (Lunar phase 2), third quarter to a waning crescent (Lunar phase 3), waning crescent to new moon (Lunar phase 4), new moon to waxing crescent (Lunar phase 5), waxing crescent to the first quarter (Lunar phase 6), first quarter to waxing gibbous (Lunar phase 7), and waxing gibbous to full moon (Lunar phase 8).

The weather parameters were obtained from two different data repositories. The respective values for the variables ‘Mean temperature’ (°C), ‘Minimum temperature’ (°C), ‘Maximum temperature’ (°C), ‘Precipitation’ (L/m^2^), ‘Mean wind speed’ (m/s), ‘Maximum wind gust’ (m/s), ‘Direction of the maximum wind gust’ (tens of degrees), ‘Hours of sunshine’ (h), ‘Mean atmospheric pressure’ (hPa), ‘Minimum atmospheric pressure’ (hPa), and ‘Maximum atmospheric pressure’ (hPa) were acquired from the official, historical data repository of the meteorological station located near to the study site (https://datosclima.es/ accessed on 2 December 2022; ID of the weather station: C249I; Coordinates: 28°26′41″ N–13°51′47″ W). ‘Relative humidity’ (%) and ‘Mean visibility’ (km) were computed from an additional free repository for the same weather local station (https://www.tutiempo.net/ accessed on 2 December 2022).

### 2.4. Statistical Analyses

First, we tested for significant differences in (1) annual variation in proportions of spontaneous onset of parturition across lunar phases, (2) the total number of male and female calves that were born in each lunar phase along the whole study period, (3) annual mean values per weather variable considering all the spontaneous onsets of labor produced in a year, and (4) mean values for each weather variable at the onset of labor depending on the sex of offspring during the whole study period, in order to have a general view of the reproductive phenology across the lunar cycle and weather on the studied population. Since all the data sets violated the normality assumption (Shapiro–Wilk test, *p* < 0.05), non-parametric Mann–Whitney U tests [57] were performed to investigate whether there were statistically significant differences in the median between associated subgroups values.

Afterward, a binary logistic regression model was developed to find the most parsimonious set of predictors that are most effective in predicting the probability for a gravid she-dromedary to give birth to a male or a female calf (dependent, binary variable) [58]. To ensure the independence of the regressors, a preliminary test of multicollinearity was run. The thumb rules used for the interpretation of the existence of multicollinearity were values of tolerance lesser than 0.10 and/or a variance inflation factor (VIF) greater than 5 [59].

Hence, the probability of giving birth to a male or a female calf was modeled according to the equation: Logit (*P*) = ln (*P*/(1 − *P*)) = *f*, where *P* is the probability of the response variable to be modeled, and *f* is defined by the following function *f* = b_0_ + b_1_ F_1_ + b_2_ F_2_ + … + b_n_ F_n_ + b_12_ F_1_ F_2_ + … + b_n−1,n_ F_n−1_ F_n_; in which b refers parameters to be fitted, and F_n_ represents the used finally as predictors in the model after the multicollinearity analysis. For the particular case of the independent predictor variable ‘Lunar phase’, since it is a categorical variable, the statistical procedure recoded it automatically.

The ‘Enter’ method was selected to include the predictive factors in the model. Several predictive performance indexes and the goodness of fit of the statistical model were computed: (i) omnibus test, (ii) coefficient of determination Nagelkerke R^2^, and (iii) Hosmer-Lemeshow (HL) statistic.

All the statistical procedures described were fitted in IBM^®^ SPSS^®^ Statistics 25.0 software (IBM SPSS, Inc., Chicago, IL, USA), with a confidence interval and level of significance of 95% and *p* < 0.05, respectively.

## 3. Results

### 3.1. Quantitative Distribution of Spontaneous Onset of Labor across Lunar Phases

No statistically significant differences (*p* > 0.05) were found neither for the annual proportions of spontaneous onset of parturition across lunar phases between years nor for the total number of male and female calves that were born in each lunar phase along the whole study period.

Figure 1 depicts the relative frequency of the spontaneous onset of labor in each lunar phase per year. An incipient change in the relative distribution of the onset of parturition across lunar phases per year can be truly appreciated more remarkably since 2004; from the concentration of most of the annual deliveries in a reduced number of lunar phases (Lunar phases 4, 5, and 6) during the first decade, the initiation of parturition in Canarian dromedaries has become an event that is distributed in a relatively homogeneous proportion between lunar phases but still with a comparatively superior predominance of the same lunar phases.

The total number of male and female calves that were born per lunar phase during the study period is represented in Figure 2. For newborn male calves, the onset of labor is most likely to be produced during the lunar phases 4 or 6 (decreasing and increasing moonlight intensity, respectively), while the lunar phase 5 (lowest moonlight intensity) is the time of the lunar calendar at which the probability of a female calf to be born is higher.

### 3.2. Mean Climate per Spontaneous Onset of Parturition

The interannual variation in the mean values for weather variables at the onsets of parturition registered each year is graphically displayed in Figure 3. Although no statistically significant differences (*p* > 0.05) were found between annual mean values per weather variable considering all the spontaneous onsets of labor produced in a year, particular trends of variation can be observed for some variables. Those weather conditions directly related to temperature (‘Mean temperature’, ‘Minimum temperature’, and ‘Maximum temperature’) and ambient air (‘Mean wind speed’, ‘Maximum wind gust’, and ‘Direction of the maximum wind gust’) have experimented with a progressive increase over the study period. The inverse tendency can be observed for those weather variables that refer to atmospheric pressure (‘Mean atmospheric pressure’, ‘Minimum atmospheric pressure’, ‘Maximum atmospheric pressure’), relative humidity, and mean visibility. The remaining studied variables (‘Precipitation’ and ‘Hours of sunshine’) showed relatively constant variation across years.

When considering the mean values for weather variables at the onsets of parturition depending on the sex of offspring through the whole study period, the differences were not statistically significant (*p* > 0.05). Slight differences did exist for the variables related to precipitation, ambient air, and humidity (Figure 4).

### 3.3. Prediction of Spontaneous Onset of Parturition Based on Environmental Variables

#### 3.3.1. Multicollinearity Analysis

After multicollinearity analyses, only the variables of ‘Lunar phase’, ‘Precipitation’, ‘Mean wind speed’, ‘Maximum wind gust’, ‘Direction of the maximum wind gust’, ‘Hours of sunshine’, ‘Relative humidity’, and ‘Mean visibility’ were retained in the binary logistic regression model (tolerance values > 0.10 and VIF values < 5; Table 1).

#### 3.3.2. Model Performance

Table 2 summarizes the evaluation of the performance of the model and the regression coefficients, Wald values, statistical significances, and odds ratio (OR) for each variable that contributes to the model. Note that the reference or baseline category for the categorical predictor ‘Lunar phase’ that is used to compare one situation to another against the dependent variable is the last phase (‘Lunar phase 8’). Since all odds ratios will be a comparison to the reference category, it means the OR for the reference category is equal to 1 [60].

## 4. Discussion

The genus *Camelus* is widely recognized to have experimented with particular adaptations in its anatomic structure, physiological functioning, and ecological niche in order to survive harsh extreme environmental conditions while maintaining the energy reserves and structural integrity considerably well balanced [61]. For a deeper understanding of the interplay between homeostasis and arid and semi-arid environments, it is useful to inquire into the particular influence of the external environment on high-energy demanding and challenging processes such as parturition and offspring care. Both the biochemical and endocrinological preparation for parturition and the posterior implication of parents to care for their offspring are subject to variation because of species-specific evolved responses to environmental selection pressures [62]. In the present study, the effects of the lunar cycle and meteorological conditions on the spontaneous onset of labor, with special attention to the sex distribution of the resulting offspring, in dromedary camels have been examined for the first time.

Although no significant variation was encountered for the temporal distribution of the onset of parturition across lunar phases and weather conditions over the years, quantitative differences can be effectively observed (Figure 1, Figure 2, Figure 3 and Figure 4). This suggests a progressive and successful adaptation of dromedaries to avoid important energy unbalance and thus minimize potential early undesirable adaptation processes that may have occurred [63].

Additionally, the variability explained by the statistical model tested is not very high (Nagelkerke R^2^; Table 2), which may lay on the fact that the impact of the environmental variables considered on the study population is not high or may rather involve a greater variety of factors nor has it been significant over time.

These findings highlight the ability of camels to switch their thermoregulation from endothermic homeothermy to becoming heterothermic and effectively minimize the impact of the external environment [45]. From the viewpoint of animal welfare, an applied topic that is growing in a contemporaneous scene but still remains limited for camels [50], it could be inferred the relatively higher importance of animal and herd-level indicators to rely on when designing camel rearing and handling protocols. That is, substantial attention should be paid to animal and herd-based measures so that the general fitness of the animals is the best possible to ensure good adaptive plasticity by putting into action their intrinsic mechanisms of adaptation to heterogeneous external environments.

Despite the moderate performance of the regression model, some of the potential predictors included did have a non-negligible effect (Table 2). Concretely, the variables that would significantly serve to predict, with a moderate accuracy rate (≈60%), the probability for a gravid she-camel to give birth to a male or a female calf were those related to the lunar phase (new moon) and ambient air (mean wind speed and maximum wind gust) (Table 2). That is, the local variation in these environmental features would have driven the adaptation of the time of delivery depending on the sex of the offspring. The phase of the new moon had the greatest influence on the model response variable (OR = 2.783; Table 2), with an OR value of 1.303 and 0.838 for the mean wind speed and maximum wind gust, respectively.

From an animal functional perspective, the underlying components of this adaptive strategy can be categorized into the differential energy costs of gestating a male or a female calf and the benefits of social cohabitation and interaction. Further, on the basis that camels have been involved in migration activities since their early domestication, these animals can be anticipated to have developed patterns of recognition of some celestial indicators. In these regards, some authors agree that the evolution of strategies to best adapt reproduction timing in dynamic environments will result in the benefit of offspring fitness both at the moment of birth and at subsequent life stages [64,65,66]. Similarly, Havenith [67] stated that behavioral modulation is the most powerful thermoregulatory effector, a condition of paramount importance in desert-living animals.

In a transnational scenario, if we replicate the methodology for other camel populations, different values for the regression model performance as well as various weather conditions with predictive potential, would be revealed. Such variance might be ascribed to the phenotypic plasticity that exists between populations due to the particularities of the socio-geographical context in which they develop (evolutionary history) [68], and that may be determining the latitudinal clines in gene frequencies of the circadian clocks in these species which are imprinting an evolutionary response. In any case, the sample size needs to be carefully considered when interpreting and comparing the statistical results since some locally adapted camel populations are reared in a restricted habitat and have a reduced effective population size [69]. However, it can be hypothesized that this between-population differentiation is not large and statistically significant given the already mentioned low genomic variability that is intrinsic to this animal species (genetic history), as well as the relatively limited eco-geographic distribution range of dromedary camels [70].

### 4.1. Low Moonlight Increases Affiliative Behavior and Territorial Defense

The relative frequency of the spontaneous onset of labor across lunar phases over the study period did experiment with a change in its quantitative trend (Figure 1). During the first decade, most of the annual deliveries were concentrated in the lunar phases of waning crescent to new moon (Lunar phase 4), new moon to waxing crescent (Lunar phase 5), and waxing crescent to the first quarter (Lunar phase 6). However, a relatively homogeneous proportion of the onset of labor events between lunar phases from the beginning of the 2000s until the present can be noticed. However, a comparatively superior predominance of the aforementioned lunar phases is still patent.

This finding could be indicative of a progressive adaptation of the animals to an environment with more controlled conditions, thus with a reduced impact of potential external stressors. Furthermore, in line with the social character of camels, the increase in the effective census of the population in recent years [54] may promote the positive evolution of spatial cohesivity and cooperation among congeners towards the group defense and benefit. In addition, the fact that births continue to be numerically more probable under certain environmental lighting conditions, even though camels are in more controlled housing conditions, could reflect the existence of a phenotypic trait with an inheritable molecular signature in these animals.

In this paradigm, Goswami, et al. [71] underlined the reasons that are responsible for the major variation in the timing and tempo of evolutionary change in mammals. Among the factors that best predicted the response of mammalian species to rapid changes in their environment were habitat, social behavior, diet, parental care, and patterns of locomotor activity. Concretely, social skills, diet, and locomotion are the most influential factors, as they are the main drivers of the morpho-functional evolution of brain size and cognitive abilities. Hence, social species, which can generally be differentiated by the development of characteristic morphological features that serve them for fighting and social display, evolve much faster than solitary species. Likewise, herbivores evolve faster than carnivores, probably because they follow changes in plants and the environment more closely than carnivores. On the other hand, species with a strict activity pattern, whether diurnal or nocturnal, would evolve more slowly than animals without fixed habits of activity.

Within this contextualization and specifically concerning the specific influence of environmental light on social behavior, an increase in the social communication and/or interaction between congeners in dim light conditions for some mammal and bird species is reported [72,73,74,75,76]. This, in turn, might enforce social cohesion both at a horizontal and vertical level. Similarly, the activity in mammals [77,78,79,80], birds [81,82], reptiles [83,84], urodeles [85], and invertebrates [86,87,88] is positively correlated with decreasing moonlight. During a full moon, animals’ visual skills may be better and move at lesser proportions since they can identify both surrounding resources and potential threats more easily, but they are also more exposed to predators in these bright light conditions. On the other hand, during dark nights, visibility is worse, and they would need to move more to access resources but are more aggregated and in continuous communication with their congeners to feel guarded or safe.

Spawning behavior in marine organisms [66,89] and amphibians [90] and spontaneous birth in cattle [16,24,91] are also generally more prevalent around the new moon. The most common hypothesis is that offspring born near the new moon tended to have a stronger fitness through to adulthood, and also as an anti-predation strategy since an increase in predation risk is patent when the full moon’s light is particularly bright [92]. In fact, in ewes, the incidence of parturition reached its peak at the full moon, but the mortality rate of offspring was also the highest at this lunar phase [93].

In these regards, a recent study provided insight into the neuroendocrine basis of mammalian aggression by revealing the role of melatonin, which is synthesized during ambient darkness, in the reduction of neurosteroid levels and the elevation of aggressive behavior [94]. Strongly linked to this, the blood circulation levels of a neurohormone such as melatonin are expected to be higher when the Earth’s electromagnetic intensity decreases from the third quarter (Lunar phase 3) to the first quarter (Lunar phase 7) [95,96].

Therefore, the social character of camels [53] could be responsible for female dromedaries’ advantage of increased social cohesion during low moonlight conditions, mediated both by hormonal signals and compensatory mechanisms, to give birth in a safe and cooperative environment. This way, not only the group performance in the struggle against external agents will be improved, but also the fact of being born into groups of cooperative and vigilant mothers could increase the altruistic behavior of females to nurse a non-filial offspring (allonursing) and reduce the thermoregulatory demands [6]. That is, camels may adapt the time of delivery to make long-term survival probability increase by sharing resources with conspecifics so that physiological resources are liberated for other processes.

Regarding the proportion of newborns of each sex that were born in a certain lunar phase throughout the entire study period, male offspring are usually born under lighter conditions (Lunar phases 4 and 6) than female calves (Lunar phase 5). This observation would be explained based on specific strategies of female dromedaries for the promotion of the survival of the offspring based on their sex, taking into consideration that survival, growth, immunity, and organ development in mammals species are higher for female newborns than their male counterparts at neonatal stages [97,98].

Then, neonatal intensive care procedures are particularly decisive for compromised newborns of the male sex, but also, they are more vulnerable to variation in parental condition since males require higher nutritional investment due to their generally larger body size [41]. Experimental research on this topic has concluded that such sex-specific differences in newborn animals can be attributed to variations in hormonal signaling and stress responses, partially due to maternal influences at organizational effects on offspring development [99,100].

Specifically, Hammadi, et al. [101] studied maternal and neonatal behaviors in dromedaries and concluded that male newborns took more time to raise their heads, stand up, and suckle their mothers. Hence, the ambient light when male calves are born, although minimal to avoid potential external threats and benefit from the greater social communication, slightly higher than the lunar phase in which the birth of female offspring is concentrated, benefits the mothers to be able to better interact visually with the immature male newborn in the first stages of its extra-uterine life. In fact, a male-biased maternal investment is reported for camels [102], as well as for other desert-living species such as the Dorcas gazelle [103] and some other ungulates [104,105,106,107,108,109].

### 4.2. Dissimilar but Complementary Effects of Air Velocity on Thermal Comfort and Olfactory Maternal Behavior

Patterns of genetic selection consistent with the adaptation of camels to desert conditions have been identified. They include tolerance to extreme temperature, dehydration, and sandy terrains [110]. Regardless of this widespread knowledge of the interaction between camels and these environmental variables, the variables related to temperature were deemed redundant in the present study (Table 1), which could be explained on the basis that local dromedaries have evolved in a relatively uniform climate due to the geographical situation of the Canary Islands [111]. Contrastingly, those climatic variables related to air velocity would serve to explain a local adaptive mechanism of Canarian dromedaries, as their interannual variation in the study area is recognized to be of notable magnitude [112].

Existing literature dealing with other animal species suggests that the variation in temperature and barometric pressure is related to the rate of birth in cows [24] but negatively associated with the occurrence of preterm calving in the same livestock species [23]. Other authors found that warm, dry, and calm conditions lead to earlier parturition dates and advanced juvenile development, whilst cold, wet, and windy weather delays birth timing and juvenile growth in bats [21]. Changes in breeding phenology and reproductive performance in response to temperature and rainfall are also reported in wild deer [113,114] and seabirds [115].

The potential of the mean wind speed and a maximum wind gust to predict the onset of parturition in Canarian dromedaries depending on the sex of offspring (Table 2) may reveal the specific influence of these weather conditions on the thermal comfort and metabolic economy of the gravid she-camels, which are expected to be more energetically compromised when the offspring that they are expecting is a male.

According to Virens and Cree [116], the heat transfer between an animal and its environment and, thus, the individual energy budget is predominantly modulated by the wind. Specifically, thermal comfort decreases as air velocity does [117], resulting in an increase in the rate of water loss that provides no cooling [118].

Under this theoretical prospect, pregnant females would initiate the delivery of their male offspring to avoid major imbalances which could put their health and that of their offspring at risk. Furthermore, a comparatively superior energy-demanding process when the velocity of ambient air is lower, and its transient increase (gust) is higher (Figure 4). Under these conditions, the temperature regulation by regulatory changes in metabolic heat production and/or evaporative heat loss is consistent; hence the metabolic economy and general organic status are compromised. These results contrast those reported by Roche, et al. [119], who described that a calf is more likely to be male than to be a female if it is born following periods of high air temperature and or high evaporation in dairy cattle. Species-specific metabolic and behavioral responses to environmental cues may explain such contrasting results.

Parallelly, according to literature, wind effects on the success at the establishment of the mother-calf bond could be expected, considering that olfactory stimuli (sniffing) are of special interest to favor mother–young interaction and fast winds impede proper olfaction [120]. Canarian she-dromedaries may be rather prone to deliver male newborns on days with decreasing wind speed given the more precocious stage of development and, thus, the need for paternal care/recognition of male neonates in comparison to female calves. In fact, the multisensory recognition through vocal and chemical communication between mother and young from parturition to weaning is reported to be more practiced towards newborns of the male sex in camels [101], beef cattle [121], and rodents [122], when compared to females.

In the context of climate change, and in view of the increasing trend of variables related to wind in the study area, future applied studies should evaluate the possible effects of this specific interannual variability on maternal rejection rates as well as abortions, preterm births, and other adverse pregnancy outcomes. The study of animal–environment interaction in local domestic scenarios will also aid in the optimum definition of handling protocols for the effective and safe management of animals.

## 5. Conclusions

New moon and ambient air have a non-negligible effect on the prediction of the onset of spontaneous parturition depending on the sex of offspring in dromedaries. These lunar and weather phenologies can be interpreted as a manifestation of phenotypic adaptation that involves species-specific social ecology and body-fluid balance features to maximize parent fitness and offspring survival. To give birth during low moonlight conditions means a safer and more cooperative environment due to the increased social cohesion between congeners, mostly regulated through hormonal pathways. Male offspring are usually born under lighter conditions than their female counterparts, which may be explained on the basis that slightly higher moonlight may benefit the mothers to be able to better interact visually with the immature male newborn in the first stages of its extra-uterine life. Pregnant she-camels would have adapted the time of delivery according to the velocity of ambient air and the intensity of its transient increases, a trade-off mediated by the influence of wind in the ability of camels to maintain a thermal balance state and minimizes metabolic use, condition of paramount importance for survival at arid and semi-arid habitats. When the mean wind speed is low, and the wind gust is high, a calf is more likely to be male. At these specific climatic conditions, the thermal comfort is negatively affected so that the gravid female would initiate the parturition to avoid major organic imbalances that could limit its posterior performance and survival, but also to interact through olfactory stimuli with the immature male newborn more accurately.

## Figures and Tables

**Figure 1 biology-12-00607-f001:**
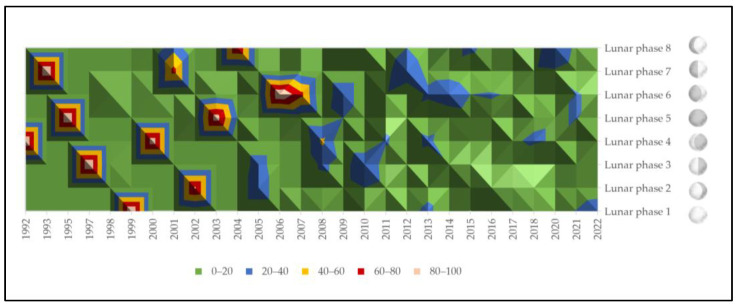
Relative frequency (percentage, %) of spontaneous onset of labor in each lunar phase per year. The relative frequency of spontaneous onset of parturition is defined as the number of births in each lunar phase in relation to all episodes of spontaneous onset of labor recorded in a specific year.

**Figure 2 biology-12-00607-f002:**
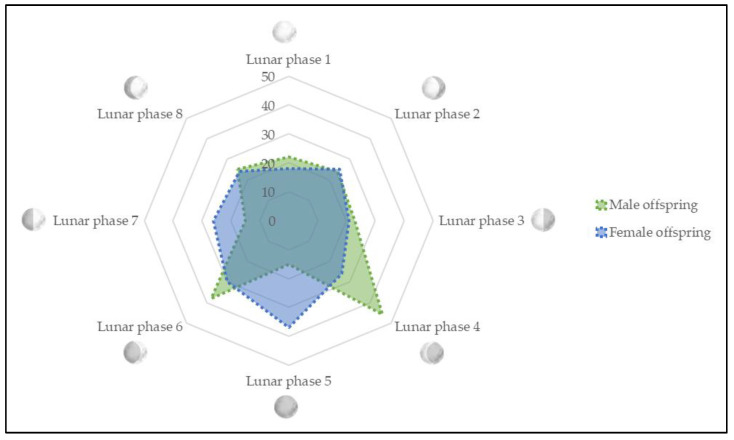
Total number of male and female calves that were born in each lunar phase throughout the whole study period.

**Figure 3 biology-12-00607-f003:**
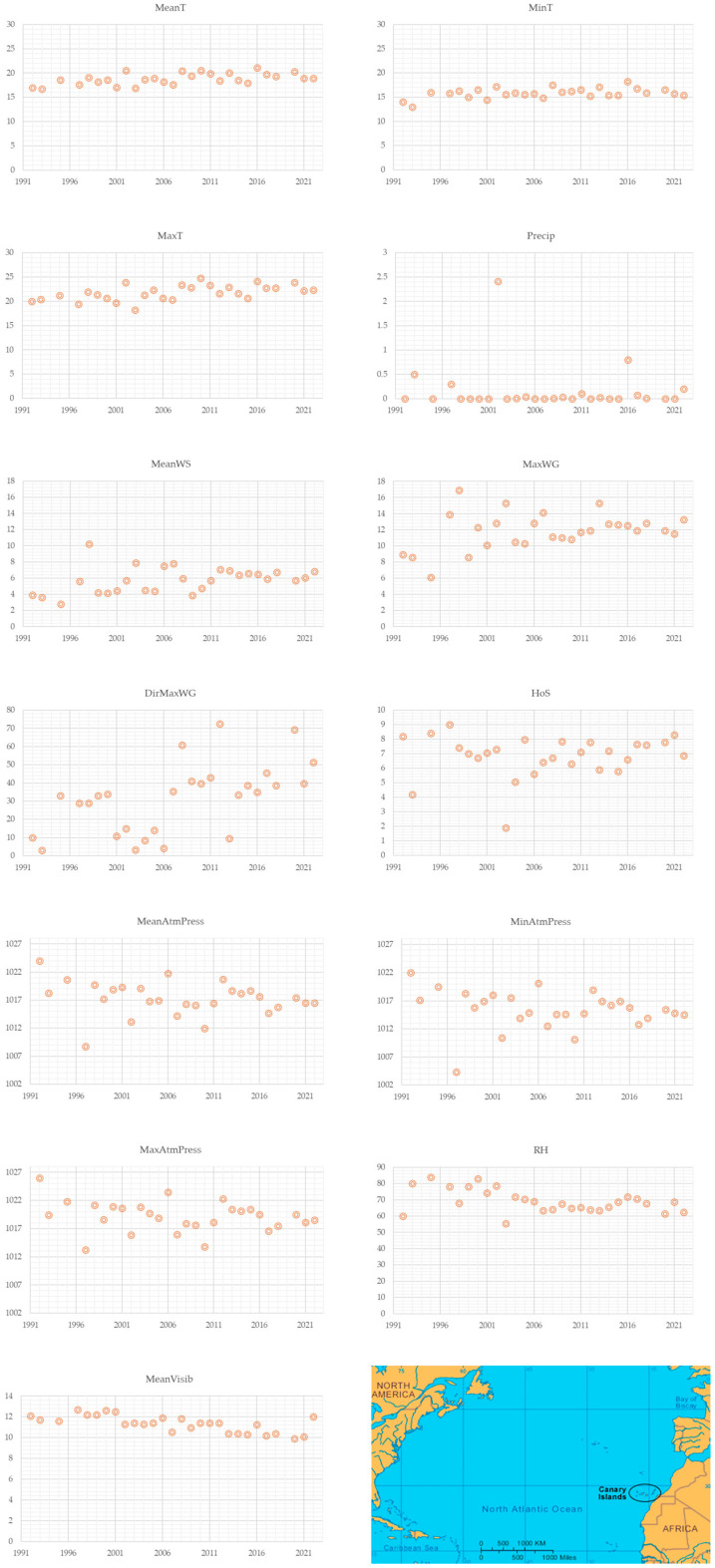
Interannual variation in the mean values for weather variables at the onsets of parturition registered each year. A map with the geographical location of the study area (black circle) is included at the bottom right of the figure. MeanT: Mean temperature, MinT: Minimum temperature, MaxT: Maximum temperature, Precip: Precipitation, MeanWS: Mean wind speed, MaxWG: Maximum wind gust, DirMaxWG: Direction of the maximum wind gust, HoS: Hours of sunshine, MeanAtmPress: Mean atmospheric pressure, MinAtmPress: Minimum atmospheric pressure, MaxAtmPress: Maximum atmospheric pressure, RH: Relative humidity, MeanVisib: Mean visibility.

**Figure 4 biology-12-00607-f004:**
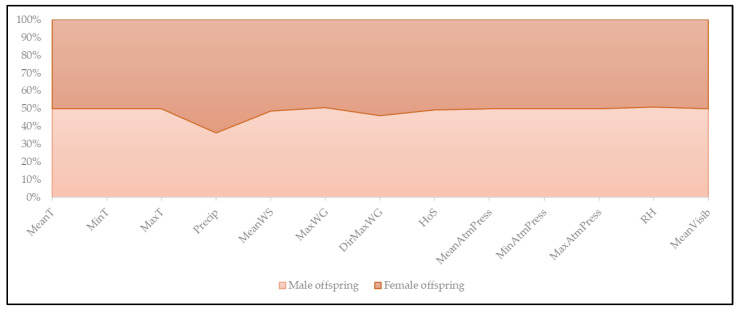
One hundred percent stacked area graph for the comparison of mean values for weather variables at the onsets of parturition depending on the sex of offspring over the study period. MeanT: Mean temperature, MinT: Minimum temperature, MaxT: Maximum temperature, Precip: Precipitation, MeanWS: Mean wind speed, MaxWG: Maximum wind gust, DirMaxWG: Direction of the maximum wind gust, HoS: Hours of sunshine, MeanAtmPress: Mean atmospheric pressure, MinAtmPress: Minimum atmospheric pressure, MaxAtmPress: Maximum atmospheric pressure, RH: Relative humidity, MeanVisib: Mean visibility.

**Table 1 biology-12-00607-t001:** Multicollinearity analysis of lunar phase and weather-related variables to discard redundant factors.

Variable	Tolerance (1 − R^2^)	VIF (1/Tolerance)
Lunar phase	0.931	1.074
Mean temperature	0.000	4575.800
Minimum temperature	0.001	1132.490
Maximum temperature	0.001	1697.666
Precipitation	0.879	1.138
Mean wind speed	0.289	3.463
Maximum wind gust	0.314	3.182
Direction of the maximum wind gust	0.846	1.183
Hours of sunshine	0.691	1.448
Mean atmospheric pressure	0.060	16.239
Minimum atmospheric pressure	0.059	16.820
Maximum atmospheric pressure	0.063	15.838
Relative humidity	0.731	1.368
Mean visibility	0.888	1.126

**Table 2 biology-12-00607-t002:** Model goodness-of-fit summary statistics and coefficient values for each independent factor in the equation.

Omnibus Test	Nagelkerke R^2^	HL Goodness-of-Fit Test	Prediction Accuracy Rate (%)	Variable	Coefficient (*B*)	Standard Error of *B*	Wald Statistic	Statistical Significance	ESTIMATED Odds Ratio (exp(*B*))
0.000	0.125	0.025	59.3	Lunar phase 8 (reference category)			17.860	0.013	1
				Lunar phase 1	−0.404	0.451	0.802	0.370	0.668
				Lunar phase 2	0.093	0.425	0.048	0.827	1.097
				Lunar phase 3	0.013	0.434	0.001	0.977	1.013
				Lunar phase 4	−0.526	0.405	1.685	0.194	0.591
				Lunar phase 5	1.024	0.439	5.430	0.020	2.783
				Lunar phase 6	−0.277	0.396	0.329	0.566	0.797
				Lunar phase 7	0.394	0.455	0.749	0.387	1.482
				Precipitation	0.156	0.106	2.149	0.143	1.168
				Mean wind speed	0.265	0.075	12.574	0.000	1.303
				Maximum wind gust	−0.177	0.056	10.123	0.001	0.838
				Direction of the maximum wind gust	0.002	0.003	0.454	0.500	1.002
				Hours of sunshine	0.051	0.043	1.371	0.242	1.052
				Relative humidity	−0.012	0.011	1.212	0.271	0.988
				Mean visibility	0.145	0.104	1.959	0.162	1.156
				Constant	−0.654	1.627	0.162	0.687	0.520

## Data Availability

Data will be made available from corresponding authors upon reasonable request.

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
