# Peer review of "Lunar Cycle, Climate, and Onset of Parturition in Domestic Dromedary Camels: Implications of Species-Specific Metabolic Economy and Social Ecology"

_biology, 2023, doi:10.3390/biology12040607_

Round 1

Reviewer 1 Report

Major comments:

The present work subjected to peer review titled: New moon and wind drive adaptive variation at onset of parturition in dromedary camels…is not an ordinary scientific study. It is mainly based on some traditional assumptions concerning the parturition onset process in the dromedary camel. In fact, it consists of a retrospective data collection of parturition onset timing and of corresponding lunar phases and weather parameters data. According to the authors, the objective of this study was to investigate the impact of lunar cycle and weather conditions on the spontaneous parturition onset of both male and female offsprings in Canarian dromedary camels. To achieve this purpose, the authors used all collected data and applied different statistical tests to verify the significance of the impact of the environmental conditions on the recorded parturition onsets. The related statistical analyses revealed no significant differences across the parturition onsets and the lunar phases as well as the weather parameters. Despite this negative result, the authors tried other tests to verify eventual consistent impact (correlation) of individual lunar phases and weather parameters with the parturition onsets. Some perceptible influences have been quoted for lunar phases, precipitation and wind variables on the parturition onsets. In my opinion, these results are scientifically consistent; first because no significant difference has been found and secondly the parturition onset process involves internal multi-endocrine control much more reliable than external environmental cues.

As a whole, the manuscript is conveniently presented and written, although some minor rectifications should be included.

The abstract does not reflect the obtained results. No indication has been given concerning the non significant effect of the indicators studied. Whilst, some declarations are just off topic (“Model performance indexes highlighted the heterothermic character of camels to greatly minimize the impact of the external environment….”). Moreover, the key words do not reflect the content of this work; they should include “parturition onset, offspring, lunar phases and weather factors” instead of “photoperiodism hormonal fulcrum homeostasis fitness; tradeoff and evolutionary response”.

The introduction is, somehow, too long. A large part has been dedicated to photoperiod and circardian rhythms; while less interest has been reserved to physiological, particularly endocrine, initiator factors of parturition onset. The theme justifications are not fully persuasive.

Material & Methods:

This part is well carried out and largely accurate concerning the population of She-camel used in this study, as well as the data collected in each case of parturition and the corresponding data of lunar cycle and weather parameters.

The used statistical tests were diverse and sufficiently defined.

Results:

The presented results issued from comparing the parturition onsets to lunar phases and to weather variables were clearly presented and illustrated. No significant differences have been found for both comparisons. Despite this fact, the authors used unnecessary other tests to investigate eventual specific correlation. However, the scientific value of these findings remains debatable.

The Discussion part is rich, diverse and constructive. However, the authors ignore the insignificance of the obtained results, but discuss, instead, the eventual relations of parturition onset with different lunar phases and weather parameters. Moreover, specific interest was given to thermoregulatory and fluid balance processes in this species; while, in the study location, there was no excessive heat and water was thoroughly available.

The conclusions are somehow speculative and subjective since there was no significance for the main results. The given conclusions were based on the collateral interrelation between some lunar phases and weather variable. These conclusions do not reflect the natural environment of this species where the conditions are different from the captive state.

The literature references are abundant and mostly recent.

Minor comments:

These comments are quoted in the enclosed version of the manuscript.(see attached file)

Additionally, some inquiries should be clarified:

-        Camel is a seasonal reproductive species; therefore, the parturitions should be concentrated in specific period of the year; which was not emphasized in the present work.

-        During different lunar phases, is it supposed that the sky was totally clear; what about cloudy nights? Meanwhile, to illustrate different lunar phases, I suggest using small graphics starting from crescent to full moon.

Author Response

Reviewer 1

Major comments:

The present work subjected to peer review titled: New moon and wind drive adaptive variation at onset of parturition in dromedary camels…is not an ordinary scientific study. It is mainly based on some traditional assumptions concerning the parturition onset process in the dromedary camel. In fact, it consists of a retrospective data collection of parturition onset timing and of corresponding lunar phases and weather parameters data. According to the authors, the objective of this study was to investigate the impact of lunar cycle and weather conditions on the spontaneous parturition onset of both male and female offsprings in Canarian dromedary camels. To achieve this purpose, the authors used all collected data and applied different statistical tests to verify the significance of the impact of the environmental conditions on the recorded parturition onsets. The related statistical analyses revealed no significant differences across the parturition onsets and the lunar phases as well as the weather parameters. Despite this negative result, the authors tried other tests to verify eventual consistent impact (correlation) of individual lunar phases and weather parameters with the parturition onsets. Some perceptible influences have been quoted for lunar phases, precipitation and wind variables on the parturition onsets. In my opinion, these results are scientifically consistent; first because no significant difference has been found and secondly the parturition onset process involves internal multi-endocrine control much more reliable than external environmental cues.

As a whole, the manuscript is conveniently presented and written, although some minor rectifications should be included.

Response: We thank the reviewer for his/her nice comments and introduction.

The abstract does not reflect the obtained results. No indication has been given concerning the non significant effect of the indicators studied. Whilst, some declarations are just off topic (“Model performance indexes highlighted the heterothermic character of camels to greatly minimize the impact of the external environment….”).

Response: We followed the reviewer’s suggestion. A specific sentence to remarks the results of the statistical analysis has been included in the abstract as requested.

Moreover, the key words do not reflect the content of this work; they should include “parturition onset, offspring, lunar phases and weather factors” instead of “photoperiodism hormonal fulcrum homeostasis fitness; tradeoff and evolutionary response”.

Response: We followed the reviewer’s suggestion. The keywords ‘photoperiodism’, ‘hormonal fulcrum’, ‘homeostasis’, ‘fitness’, ‘tradeoff’ and ‘evolutionary response’ have been deleted, and the keywords ‘parturition onset’, ‘lunar phase’, ‘weather factors’ and ‘heterothermia’ included.

The introduction is, somehow, too long. A large part has been dedicated to photoperiod and circadian rhythms; while less interest has been reserved to physiological, particularly endocrine, initiator factors of parturition onset. The theme justifications are not fully persuasive.

Response: Introduction has been summarized. In addition, we have included details on the different factors (endocrine, nervous and immune system) that lead to the initiation of parturition. Moreover, the seasonality at camel reproduction as well as the larger dataset on climate related variables included in the present research when compared to previous research in the same specific field, have been emphasized. Indeed, the weather-related variables such as temperature, relative humidity and hours of sunshine, which are the most commonly included ones in those research studies aiming at studying the relationship between environmental stressors and reproductive function in animal species, have no significant effect in the present study.

Material & Methods:

This part is well carried out and largely accurate concerning the population of She-camel used in this study, as well as the data collected in each case of parturition and the corresponding data of lunar cycle and weather parameters.

The used statistical tests were diverse and sufficiently defined.

Response: We thank the reviewer for his/her nice comments.

Results:

The presented results issued from comparing the parturition onsets to lunar phases and to weather variables were clearly presented and illustrated. No significant differences have been found for both comparisons. Despite this fact, the authors used unnecessary other tests to investigate eventual specific correlation. However, the scientific value of these findings remains debatable.

Response: We understand the reviewer concern. However, the rationale behind the tests used is as follows: First, we tested for significant differences in (1) annual variation in proportions of spontaneous onset of parturition across lunar phases, (2) total number of male and female calves that were born in each lunar phase along the whole study period, (3) annual mean values per weather variable considering all the spontaneous onsets of labour produced in a year, and (4) mean values for each weather variable at onset of labour depending on the sex of offspring during the whole study period, in order to have a general view of the reproductive phenology across the lunar cycle and weather on the studied population. This univariate analysis is intended to study the relationship between two variables.

Afterwards, a binary logistic regression model was developed to find the most parsimonious set of predictors that are most effective in predicting the probability for a gravid she-camel to give birth to a male or a female calf (dependent, binary variable). This multivariate analysis not only shows whether there is or not a significant association between a binary variable A (sex of offspring) and a set of independent variables, but also is able to compute the expected likelihood of the event recorded in variable A considering the statistical relationship (interaction effects) between several independent variables.

In summary, a combination of univariate and multivariate methods should be used to unveil information in biological phenomena studies. In general, multivariate methods focus on the relations between factors and their orchestrated or complementary behavior in relation to biological processes, and univariate methods on independent changes in the levels of single factors. In any case, it is crucial to be aware that the results of univariate and multivariate analyses do not necessarily coincide but provide complementary information [1]. These results should be interpreted within the statistical framework (uni or multivariate) with which they have been produced, and it is advisable not to seek validation of univariate results by means of multivariate analysis and viceversa [2,3].

The Discussion part is rich, diverse and constructive. However, the authors ignore the insignificance of the obtained results, but discuss, instead, the eventual relations of parturition onset with different lunar phases and weather parameters.

Response: We thank the reviewer for his/her nice comments. We understand the reviewer concern. However, a we explained in the paper, R-squared is the percentage of the variance in the dependent variable that is predictable from the independent variables. Regarding Hosmer-Lemeshow test, this statistical test calculates if the observed event rates match the expected event rates in population subgroups.

For both statistical indexes, statistical significance is partly a function of sample size. If you have a large sample, even small effects will be significant. If you have a small sample, large effects will not be significant [4]. Many researchers turned to using effect sizes because evaluating effects using p-values alone can be misleading. But effect sizes can be misleading too if you don’t think about what they mean within the research context. In other words, depending upon your question, significance testing may be meaningful regardless of the effect size [5].

Then, a low value of R2 and HL test doesn’t mean it’s bad, unworthy of being interpreted, or useless. Small effect sizes can have scientific or clinical significance. ‘Good’ values for these measures depend on the field, which in turn drives the interpretation of the results [6,7].

Specifically, in the present research, the low values of R2 and HL test are deeply interpreted in the discussion section. Both values reinforce the empirical knowledge on the heterothermy character of camels. Additionally, as a part of the interpretation of the results obtained, when looking at parameter estimates and predictions such as Odds Ratio, Wald Statistic and Statistical Significance of Predictor variables, it can be concluded that the model is useful for representing specific relationships between environmental variables such as lunar phase 8, wind speed and maximum wind gust, and onset of parturition in dromedary camels.

In summary, the model highlights the heterothermic character of camels to greatly minimize the impact of external environment, as well as the significance of some specific environment related variables to predict the onset of parturition in dromedary camels depending on the sex of offspring. In the last case, despite of the fact that ample knowledge exists on the interaction between camels and environmental variables related to temperature, these specific variables resulted redundant in the present study (Table 1). Contrastingly, our results are the first empirical research that finds a significant relationship between climatic variables related to air velocity, moon phase and reproductive physiology in this animal species.

Moreover, specific interest was given to thermoregulatory and fluid balance processes in this species; while, in the study location, there was no excessive heat and water was thoroughly available.

The variables related to temperature were deemed redundant in the present study (Table 1), which can be explained on the basis that local dromedaries have evolved in a relatively warm climate due to the geo-graphical situation of the Canary Islands [8]. Instead, those climatic variables related to air velocity would serve to explain a local adaptive mechanism of Canarian dromedaries, as their interannual variation at the study area are recognized to be of notable magnitude [9].

The conclusions are somehow speculative and subjective since there was no significance for the main results. The given conclusions were based on the collateral interrelation between some lunar phases and weather variable. These conclusions do not reflect the natural environment of this species where the conditions are different from the captive state.

Response: The conclusions do not reflect the natural environment because the studied animals are kept in a captive environment, but it does not mean that the natural physiology of the animals have completely changed because of this condition. Indeed, the results are discussed considering the potential influencing effects of a more controlled environment in the physiology and behavior of the animals, but also the existence of a phenotypic trait with an inheritable molecular signature in these animals. Anyway, we included the word domestic in the title to save the reviewer’s concern.

The literature references are abundant and mostly recent.

Minor comments:

These comments are quoted in the enclosed version of the manuscript.(see attached file)

Response: All the comments have been addressed.

Additionally, some inquiries should be clarified:

-        Camel is a seasonal reproductive species; therefore, the parturitions should be concentrated in specific period of the year; which was not emphasized in the present work.

Response: A sentence on the seasonal character of the camel reproduction has been included in the Introduction of the manuscript. However, the present research is not aimed at studying the seasonal distribution of the deliveries but at unravelling the interactive effects between environmental variables (moon phase and weather) and the onset of parturition in dromedary camels, assuming that deliveries will be concentrated in the breeding season.

-        During different lunar phases, is it supposed that the sky was totally clear; what about cloudy nights? Meanwhile, to illustrate different lunar phases, I suggest using small graphics starting from crescent to full moon.

Response: The impact of cloud cover or the fraction of the sky obscured by clouds on average when observed from a particular location, has been evaluated indirectly through the variable ‘Mean visibility’. By definition, mean visibility is the measure of the distance at which an object or light can be clearly discerned. In meteorology it depends on the transparency of the surrounding air and as such, it is unchanging, no matter what is the ambient light level or time of day. It is also influenced by weather variables such as temperature, wind speed, and precipitation [10,11].

But also the cloud cover correlates the sunshine duration (‘Hours of sunshine’) as the least cloudy localities are the sunniest ones while the cloudiest areas are the least sunny places, given clouds can block sunlight, especially on sunrise and sunset where sunlight is already limited [12,13].

Reviewer 2 Report

Dear Authors,

It,s an interesting subject, and data collection is perfect but the results were not statistically reliable. In table 2, most statistics were not appropriate. The R square of the logit model is very low. The Predictor variables are not significant (Statistical significance) except for lunar phases 8, 5, and wind. The hosmer-Lemeshow test (HL test) ( goodness of fit test) is less than 0.05. Finally, The statistical model is not statistically robust.

Author Response

Reviewer 2

It-s an interesting subject, and data collection is perfect but the results were not statistically reliable. In table 2, most statistics were not appropriate. The R square of the logit model is very low. The Predictor variables are not significant (Statistical significance) except for lunar phases 8, 5, and wind. The hosmer-Lemeshow test (HL test) ( goodness of fit test) is less than 0.05. Finally, The statistical model is not statistically robust.

Response: R-squared is the percentage of the variance in the dependent variable that is predictable from the independent variables. Regarding Hosmer-Lemeshow test, this statistical test calculates if the observed event rates match the expected event rates in population subgroups.

For both statistical indexes, statistical significance is partly a function of sample size. If you have a large sample, even small effects will be significant. If you have a small sample, large effects will not be significant [4]. Many researchers turned to using effect sizes because evaluating effects using p-values alone can be misleading. But effect sizes can be misleading too if you don’t think about what they mean within the research context. In other words, depending upon your question, significance testing may be meaningful regardless of the effect size [5].

Then, a low value of R2 and HL test doesn’t mean it’s bad, unworthy of being interpreted, or useless. Small effect sizes can have scientific or clinical significance. ‘Good’ values for these measures depend on the field, which in turn drives the interpretation of the results [6,7].

Specifically, in the present research, the low values of R2 and HL test are deeply interpreted in the discussion section. Both values reinforce the empirical knowledge on the heterothermy character of camels. Additionally, as a part of the interpretation of the results obtained, when looking at parameter estimates and predictions such as Odds Ratio, Wald Statistic and Statistical Significance of Predictor variables, it can be concluded that the model is useful for representing specific relationships between environmental variables such as lunar phase 8, wind speed and maximum wind gust, and onset of parturition in dromedary camels.

In summary, the model highlights the heterothermic character of camels to greatly minimize the impact of external environment, as well as the significance of some specific environment related variables to predict the onset of parturition in dromedary camels depending on the sex of offspring. In the last case, despite of the fact that ample knowledge exists on the interaction between camels and environmental variables related to temperature, these specific variables resulted redundant in the present study (Table 1). Contrastingly, our results are the first empirical research to find a significant relationship between climatic variables related to air velocity, moon phase and reproductive physiology in this animal species.

References

  1. Grove, H.; Jørgensen, B.M.; Jessen, F.; Søndergaard, I.; Jacobsen, S.; Hollung, K.; Indahl, U.; Færgestad, E.M. Combination of statistical approaches for analysis of 2-DE data gives complementary results. Journal of proteome research 2008, 7, 5119-5124.
  2. Mohammadi, R.; Sadeghzadeh, B.; Poursiahbidi, M.M.; Ahmadi, M.M. Integrating univariate and multivariate statistical models to investigate genotype× environment interaction in durum wheat. Annals of applied biology 2021, 178, 450-465.
  3. Saccenti, E.; Hoefsloot, H.C.; Smilde, A.K.; Westerhuis, J.A.; Hendriks, M.M. Reflections on univariate and multivariate analysis of metabolomics data. Metabolomics 2014, 10, 361-374.
  4. Liu, I.; Fernández, D. Discussion on “Assessing the goodness of fit of logistic regression models in large samples: A modification of the Hosmer‐Lemeshow test” by Giovanni Nattino, Michael L. Pennell, and Stanley Lemeshow. Biometrics 2020, 76, 564-568.
  5. Newman, I.; Newman, C. A discussion of low r-squares: Concerns and uses. Educational research quarterly 2000, 24, 3.
  6. Grace-Martin, K. Can a regression model with a small R-squared be useful. The Analysis Factor 2012.
  7. Allison, P. What’s the best R-squared for logistic regression. Statistical Horizons 2013, 13.
  8. Rancel Rodriguez, N.M. Biodiversity of epiphyllous, heterocyst-forming cyanobacteria in the laurel forest of the Canary Islands. Universität zu Köln, 2016.
  9. Azorin-Molina, C.; Menendez, M.; McVicar, T.R.; Acevedo, A.; Vicente-Serrano, S.M.; Cuevas, E.; Minola, L.; Chen, D. Wind speed variability over the Canary Islands, 1948–2014: focusing on trend differences at the land–ocean interface and below–above the trade-wind inversion layer. Climate Dynamics 2018, 50, 4061-4081.
  10. Xue, D.; Li, C.; Liu, Q. Visibility characteristics and the impacts of air pollutants and meteorological conditions over Shanghai, China. Environmental Monitoring and Assessment 2015, 187, 1-10.
  11. Pulugurtha, S.S.; Mane, A.S.; Duddu, V.R.; Godfrey, C.M. Investigating the influence of contributing factors and predicting visibility at road link-level. Heliyon 2019, 5, e02105.
  12. Essa, K.S.; Etman, S.M. On the relation between cloud cover amount and sunshine duration. Meteorology and Atmospheric Physics 2004, 87, 235-240.
  13. Matuszko, D. Influence of cloudiness on sunshine duration. International Journal of Climatology 2012, 32, 1527-1536.

Round 2

Reviewer 1 Report

The authors respond to all minor remarks and not fully to major remarks. The revised manuscript has been improved and appears more consistent.